# Common helical V1V2 conformations of HIV-1 Envelope expose the α4β7 binding site on intact virions

Constantinos Kurt Wibmer [1,2,6], Simone I. Richardson[1,2], Jason Yolitz [3,4], Claudia Cicala[3], James Arthos[3], Penny L. Moore[1,2,5] & Lynn Morris[1,2,5]

The α4β7 integrin is a non-essential HIV-1 adhesion receptor, bound by the gp120 V1V2 domain, facilitating rapid viral dissemination into gut-associated lymphoid tissues. Antibodies blocking this interaction early in infection can improve disease outcome, and V1V2-targeted antibodies were correlated with moderate efficacy reported from the RV144 HIV-1 vaccine trial. Monoclonal α4β7-blocking antibodies recognise two slightly different helical V2 conformations, and current structural data suggests their binding sites are occluded in prefusion envelope trimers. Here, we report cocrystal structures of two α4β7-blocking antibodies from an infected donor complexed with scaffolded V1V2 or V2 peptides. Both antibodies recognised the same helix-coil V2 conformation as RV144 antibody CH58, identifying a frequently sampled alternative conformation of full-length V1V2. In the context of Envelope, this α-helical form of V1V2 displays highly exposed α4β7-binding sites, potentially providing a functional role for non-native Envelope on virion or infected cell surfaces in HIV-1 dissemination, pathogenesis, and vaccine design.

[1] Centre for HIV and STIs, National Institute for Communicable Diseases (NICD), of the National Health Laboratory Service (NHLS), Johannesburg 2131, South Africa. [2] Faculty of Health Sciences, University of the Witwatersrand, Johannesburg 2000, South Africa. [3] Laboratory of Immunoregulation, National Institute of Allergy and Infectious Diseases, National Institutes of Health, Bethesda, MD 20892, USA. [4] National Institutes of Health, Institutes of Health–Johns Hopkins University Graduate Partnership Program, Bethesda, MD 20892, USA. [5] Centre for the AIDS Programme of Research in South Africa (CAPRISA), University of KwaZulu-Natal, Durban 4041, South Africa. [6] Present address: The Scripps Research Institute, La Jolla 92037, USA. Correspondence and requests for materials should be addressed to C.K.W. (email: c.k.wibmer@gmail.com)

The HIV-1 envelope glycoproteins (Env) are the only virally encoded proteins presented on virion membrane surfaces. They are expressed from the *env* gene as a single gp160 precursor, that is extensively modified by heterogeneous N-linked glycosylation and tyrosine sulphation, before being cleaved by the host protease furin[1]. The resulting gp120 and gp41 subunits remain noncovalently associated and assemble with two other cleaved gp160 protomers to form a metastable trimer of heterodimers that mediates HIV-1 entry and infection. Interprotomer interactions are found between the three gp41 molecules, as well as at the apex of gp120 where three V1V2 and V3 loop regions form the trimer association domain[2–5]. When the gp120 receptor binding subunit interacts with the host cellular receptor CD4, large conformational rearrangements reposition V1V2, expose the V3 loop along with the associated CCR5 coreceptor binding site, and free the fusion peptide, thus activating the gp41 membrane fusion subunit[6,7]. Infectious HIV-1 particles display only 10–14 properly folded and post-translationally processed, entry-competent, prefusion Env trimers per virion[8]. A portion of the remaining viral surface is coated with aberrant forms of Env, also erroneously referred to as "junk" Env, formed from uncleaved or inappropriately processed gp160 precursors, non-trimeric protomers, or Env that has prematurely sampled the CD4-bound state[9–11]. These proteins appear to act as immunological decoys, thwarting the neutralising antibody response by exposing several highly immunogenic non-neutralising antibody epitopes such as the V3 loop. However, antibodies bound to these non-neutralisable epitopes may still play an important role in countering HIV-1 infection and improving disease outcome by engaging Fc-mediated effector functions[12].

In addition to binding the entry receptors, HIV-1 also binds to target cells through various host proteins that have been incorporated onto viral surfaces to promote cell invasion and syncytium formation through colocalization of Env and the entry receptors[13–15]. The gp120 subunit has also been implicated in direct binding of other host cell surface proteins such as the dendritic cell specific intercellular adhesion molecule-3-grabbing non-integrin (DC-SIGN) or the gut mucosal homing integrin α4β7[16,17]. Although not essential for virus entry, these adhesion receptors play an important role in viral dissemination and pathogenesis[18,19]. The α4β7 integrin colocalizes with CD4, defining a highly HIV-1 susceptible T-cell population that is infected and depleted from gut-associated lymphoid tissues (GALT) within the first two weeks of infection[19,20]. The α4β7 integrin binds to a highly-conserved tripeptide motif in the V2 region of gp120 (at positions 179–181) that mimics a binding loop in its natural ligands[17]. More recently, an additional determinant of α4β7 binding has been suggested upstream of this site at V2 residues 170–173[21]. Monoclonal antibodies (mAbs) that block α4β7 binding limit the early loss of CD4 T cells, or the establishment of certain viral reservoirs. In the non-human primate model, α4β7 binding antibodies can reduce mucosal transmission from repeated low-dose challenges[22], and early co-administration of antiretroviral therapy with α4β7 binding antibodies has been shown to significantly improve disease outcome by supressing viral rebound[23]. Similarly, antibodies that bind to V1V2 were identified as a correlate of reduced infection risk in the RV144 HIV-1 vaccine trial, which showed 31% efficacy in the absence of neutralising antibodies[24]. Antibodies isolated from these RV144 vaccine recipients were shown to block the binding of V2 to α4β7. Altogether these data suggest a potentially beneficial role for the early induction of antibodies to the V2 region of HIV-1 gp120 capable of blocking α4β7 binding. However, in the context of the native, prefusion trimer, the α4β7 binding site is occluded in a 5-stranded β-barrel by intraprotomer interfaces

with gp120[25,26], and a mechanism through which such antibodies might act against intact virions is lacking.

Primary sequence elements of the V1V2 domain can be divided into interspersed conserved or variable regions. The conserved terminal strands A and D connect gp120 to V1V2, and also form part of the 'bridging sheet' that stabilises gp120 in the CD4-bound conformation. Between them, the hypervariable loop V1 (residues 132 and 156) precedes V2 and is bound at its base by a disulphide bond between C131 and C157. The subsequent N terminal region of V2 (residues 157–181) is relatively conserved and forms the primary target of all cross-reactive V2 antibodies, while the C terminal portion of V2 is a hypervariable loop (residues 182 – 189). Antibodies targeting V1V2 can be grouped into three broadly defined binding modalities[27]. The 5-stranded β-barrel form of V1V2 that is found on native, prefusion Env is bound by quaternary structure preferring V2 neutralising antibodies (V2q mAbs)[28–34], as well as by conformation dependent non-neutralising V2 antibodies that recognise the primary α4β7 integrin binding residues 179-181 in the context of non-trimeric Env (V2i mAbs)[26,27]. A third group, also comprised of non-neutralising V2 antibodies, recognises short V2 peptides (V2p mAbs)[35]. These V2p mAbs are the types of antibodies that were initially isolated from RV144 vaccine recipients and bind to an epitope that overlaps both the primary and potential secondary α4β7 binding determinants in V2. In accordance, an analysis of the Env sequences from breakthrough infections in RV144 identified two genetic signatures in V2 that were associated with vaccine efficacy at positions 169 and 181, near the secondary and primary α4β7 binding determinants, respectively[36]. Cocrystal structures of RV144 vaccine antibodies CH58 and CH59 showed how these two sites colocalise in V2p antibody paratopes[35]. However, these data also revealed an unexpected degree of conformational plasticity for V2 peptides, which exhibited helical structures that were vastly different from the 5-stranded β-barrel bound by V2q and V2i mAbs[37]. Molecular dynamics simulations, in silico modelling, and Nuclear Magnetic Resonance studies have all suggested that unconstrained V2 could adopt more helical alternative conformations similar to those observed in the CH58 and CH59 cocrystal structures[27,38]. The biological significance of helical V2 variants and whether these epitopes exist in the context of HIV-1 virions remained unclear but has important implications for understanding the role of α4β7 blocking V2 antibodies in pathogenesis and vaccination.

We recently isolated two mAbs from an HIV-1 infected donor (CAP228) that bind to V2 peptides, confirming that V2p epitopes are in fact presented during HIV-1 infection (van Eeden, C., Wibmer, C. K. et al., manuscript submitted). Here, we first determined the cocrystal structures of these new antibodies bound to V2 peptides, identifying a common class of V2p mAbs that share a conserved α-helix-coil epitope with the RV144 mAb CH58. We subsequently determined the cocrystal structure of one mAb (CAP228-16H) bound to the entire V1V2 domain, while still N and C terminally constrained as a scaffolded protein. This revealed an alternatively folded V1V2 exhibiting a highly exposed α4β7 binding site on Envelope. These data provide a potentially functional role for aberrant Env on the surface of HIV-1 virions and further elucidate a mechanism by which V2p mAbs can mediate antiviral activities by blocking the interaction with α4β7 on CD4+ T cells.

## Results

**Helix-coil V2 conformation is a reproducible antibody target**. The two monoclonal antibodies previously isolated from HIV-1 infected donor CAP228 bind V2 peptides using a signature

anionic CDR-L2 sequence (referred to as the ED- or DDxD-motifs) derived from different light chain genes, IGVλ6-57 for CAP228-3D and IGVλ3-21 for CAP228-16H (van Eeden, C., Wibmer, C. K. et al., manuscript submitted). This was similar to the RV144 vaccine elicited antibody CH58 which also uses the IGVλ6-57 gene, as well as CH59 which uses the IGVλ3-10 gene[39]. Both the CAP228 antibodies also shared the same IGHV5-51 germline heavy chain gene as CH58, suggesting that V2p antibodies might be grouped by common modes of V2 recognition. To investigate this, we screened the CAP228 antibodies for binding to various 1FD6 scaffolded V1V2 proteins and short V2 peptides (Supplementary Fig. 1A). The CAP228 mAbs bound to a diverse array of V2 antigens, facilitating cocrystal structures with various heterologous HIV-1 strains both as free peptides, or N and C terminally constrained on small protein scaffolds (Table 1). A structure of the first mAb, CAP228-16H, bound to a scaffolded V1V2 domain from a heterologous strain CAP225 was solved at 2.3 Å resolution (Table 1a and Fig. 1a), while a structure of the second mAb CAP228-3D, in complex with a V2 peptide (gp120 residues 164–182) from heterologous virus strain CAP45 was solved at 2.6 Å resolution (Table 1b and Fig. 1b). The crystal lattice formed from CAP228-16H bound to 1FD6-V1V2 was constructed predominantly by Fab-Fab interactions. As a result, only the V2p epitope in direct contact with the antibody was ordered, and the V1V2 scaffold could not be resolved (Supplementary Fig. 1B). However, the visible region included all the V2 residues also bound by CAP228-3D and CH58, allowing both CAP228 mAb complexes to be compared to the previously determined structure of CH58 bound to a V2 peptide (PDB ID: 4HPO) from the vaccine strain 92TH023 (Fig. 1c).

Remarkably, despite slight differences in $V_H/V_L$ packing and substantial sequence variation between antigens derived from genetically diverse strains (Fig. 1d-bottom), the V2p epitopes of both the CAP228 mAbs, as well as that of CH58, reproducibly adopted the same helix-coil conformation. In this IGHV5-51 bound form, V2 residues 164–176 were folded into an α-helix, while residues 177–185 formed an extended coil that was oriented by the heavy chain component of each respective paratope. When overlaid, the corresponding regions of V2 bound by each of the three IGHV5-51 antibodies had an overall Cα root-mean-square deviation (Cα RMSD) of 1.2 Å$^2$ for the CH58/CAP228-3D comparison, and 1.4 Å$^2$ for the CH58/CAP228-16H comparison, respectively (Fig. 1e). This differed significantly from the coil and $3_{10}$-helix structure recognised by the RV144 mAb CH59 (Fig. 1f), with an Cα RMSD of 3.9 Å$^2$ for the CH58/CH59 comparison), or from epitopes recognised as part of a 5-stranded β-barrel by V2i (Cα RMSD of 5.4 Å$^2$ for the CH58/830 A comparison) or V2q (Cα RMSD of 6.8 Å$^2$ for the CH58/PG9 comparison) mAbs (Fig. 1g). Thus, while the V2 region of Env possesses a high degree of sequence diversity and structural plasticity, residues 164-182 appear to reproducibly adopt an immunogenic helix-coil configuration either after vaccination (as in RV144) or during HIV-1 infection (described here).

**IGHV5-51/lambda gene pairing is preconfigured for V2 peptides.** The comparison of CAP228-16H, CAP228-3D and CH58 bound to equivalent short V2 peptides from CAP225, CAP45 or 92TH023, revealed common structural elements exploited by antibodies using the IGHV5-51 genes to interact with V2 (Fig. 2). At the elbow between the α-helix and coil segments of V2, the highly-conserved HIV-1 residues L175 and F176 (Fig. 2a) insert into a depression formed from hydrophobic/aromatic amino acids at the heavy-light chain interface (Fig. 2b). HIV-1 V2 residue 172 is also partially buried, but V172 is less conserved, with a hydrophilic glutamate side chain present in half of globally

circulating viruses (Fig. 2a, b). Specifically, the hydrophobic depression is comprised of side chains that are both conserved from the antibody germline genes (eg: CDR-H1 position W33) or conservatively substituted through somatic hypermutation (eg: CDR-H2 positions I/M50 and K/R58), but also included substantial contributions from the CDR-H3 and CDR-L3 loops (Fig. 2b, c). Overall, CAP228-16H and CAP228-3D IGHV5-51 heavy chain interactions with the CAP45 V2 peptide buried 570 Å$^2$ and 435 Å$^2$ of surface area, respectively, similar to the 596 Å$^2$ that is buried by a 92TH023 V2 peptide in the CH58 heavy chain, and included key interactions with V2 residues 173, 176, 178 and 180 (Supplementary Fig. 2A)

Like the CDR-L2 of V2p mAbs (Supplementary Fig. 2A), the CDR-H2 regions of all three IGHV5-51 V2p mAbs exhibited a conserved, negatively charged patch, created by acidic residues at positions 54 and 56 (Fig. 2c, d, outlined in yellow). These CDR-H2 residues are preconfigured in the germline to interact with the usually positively charged amino acid at position 178 in V2 (Fig. 2d). Additional affinity maturation of CAP228-16H through the D56E and K58R mutations further stabilised this interaction, by facilitating additional contacts with the V2 peptide backbone (Fig. 2d–left panel). Similarly, both CAP228-16H and CH58 acquired an arginine substitution in the CDR-H1 at position 28 that formed a key salt bridge interaction with the α4β7 binding tripeptide motif of gp120 (Fig. 2c, d, marked in green, and Supplementary Fig. 2). This mutation locks down residue D180 in the C-terminal portion of the V2 peptide, such that it's absence in the CAP228-3D mature antibody resulted in a greater degree of C-terminal V2 disorder (reflected by an ~150 Å$^2$ loss of binding surface area–Supplementary Fig. 2B) similar to what has been seen for the CH58 UCA[40]. Packing interactions in the CAP225-V1V2 crystal lattice displace the R28 side chain, but an R97 residue in the CAP228-16H CDR-H3 compensates for this by forming a salt bridge with an alternate rotamer of D180 (Fig. 2d).

In contrast to these similarities, analysis of the CDR-H3 loops from all three antibodies revealed no correlation in charge distribution, predicted D-gene usage or loop conformation (Fig. 2c, d, marked in pink), although there was a tendency for relatively longer CDR-H3 loops compared to the average human repertoire (according to the abYsis database). CH58 uses its CDR-H3 loop to form critical hydrogen bonds and aromatic stacking interactions with the V2 residue H173, such that variation away from this relatively less common immunotype partially abrogates CH58 binding, thus limiting the mAbs cross-reactivity[35]. In contrast, the CAP228 antibodies recognise more diverse immunotypes at this position, including the most globally prevalent Y173 immunotype present in the CAP45 and CAP228 strains, as well as the rarer H173 or R173 immunotypes found in the 92TH023 or CAP225 strains, respectively (Figs. 1d, 2c, d, and Supplementary Fig. 2A). In the case of CAP228-16H, this tolerance was partially due to the use of the IGVL3-21 light chain gene (not seen in RV144 vaccine elicited antibodies), that was both able to accommodate the longer Y/R173 side chains, as well as directly hydrogen bond with them via a shorter CDR-L2 D50 residue (Fig. 2d and Supplementary Fig. 2). The CAP228-16H and CAP228-3D light chains buried 374 Å$^2$ and 421 Å$^2$ of V2 surface area respectively, compared to the 537 Å$^2$ buried by CH58. CAP228-3D used the same heavy and light chain genes as CH58, but unlike CH58 its CDR-H3 was positioned away from residue 173, forming productive interactions with K169 and allowing for more diverse immunotypes at position 173 to be accommodated. These data suggest that the IGHV5-51 gene is both preconfigured to interact with V2, and poised to undergo key, relevant affinity maturation events (independent of CDR-H3 recombination). Individual IGHV5-51 heavy chain signatures, together with commonalities at the heavy-light chain interface,

**Table 1 X-ray crystallographic data and refinement statistics (molecular replacement) for the antigen binding fragments of the CAP228 antibodies in complex with various V2 antigens**

| | A<br>CAP228-16H bound to CAP225 V1V2-1FD6 | B<br>CAP228-3D bound to CAP45 V2 peptide | C<br>CAP228-16H bound to CAP228 V1V2-1FD6 |
|---|---|---|---|
| **Data collection** | | | |
| Space group | $P12_11$ | $C121$ | $C121$ |
| Cell dimensions | | | |
| $a, b, c$ (Å) | 110.35, 41.76, 144.52 | 208.36, 43.81, 119.91 | 81.99, 73.79, 191.03 |
| $\alpha, \beta, \gamma$ (°) | 90, 97, 90 | 90, 94, 90 | 90, 92, 90 |
| Resolution (Å) | 50.00–2.30 (2.34–2.30)[a] | 50.00–2.60 (2.64–2.60) | 50.00–3.21 (3.27–3.21) |
| Unique reflections | 58,982 | 33,817 | 19,842 |
| $R_{pim}$ | 0.121 (0.743) | 0.122 (0.390) | 0.098 (0.500) |
| $I/\sigma I$ | 7.6 (4.2) | 7.5 (3.9) | 6.8 (2.0) |
| $CC_{1/2}$ | 0.952 (0.560) | 0.936 (0.812) | 0.958 (0.856) |
| Completeness (%) | 99.6 (99.4) | 100.0 (100.0) | 89.7 (89.4) |
| Redundancy | 3.6 (3.6) | 6.0 (3.6) | 3.2 (3.2) |
| **Refinement** | | | |
| Resolution (Å) | 36.10–2.30 | 42.87–2.60 | 40.98–3.21 |
| No. of reflections | 58721 (4982) | 33801 (3122) | 17788 (1388) |
| $R_{work}/R_{free}$ | 0.19/0.22 | 0.22/0.25 | 0.22/0.27 |
| No. atoms (no H) | 7237 | 6694 | 7719 |
| Protein | 6792 | 6529 | 7684 |
| Ligand | 39 | 0 | 0 |
| Water | 406 | 168 | 0 |
| $B$ factors (no H) (Å²) | 48 | 64 | 99 |
| Protein | 48 | 65 | 99 |
| Ligand | 120 | 0 | 0 |
| Water | 43 | 39 | 0 |
| Wilson B-factor (Å²) | 33 | 40 | 82 |
| RMS deviations | | | |
| Bond lengths (Å) | 0.002 | 0.004 | 0.004 |
| Bond angles (°) | 0.615 | 0.596 | 0.681 |
| Ramachandran plot (%) | | | |
| Favoured regions | 98.87% | 97.83% | 95.08% |
| Outlier regions | 0.00% | 0.00% | 0.00% |
| Rotamer outliers (%) | 0.00% | 0.00% | 0.00% |
| Molprobity clash score | 0.67 | 1.56 | 3.31 |
| Molprobity overall score | 0.72 | 0.94 | 1.83 |
| **PDB ID** | 6FY2 | 6FY3 | 6FY1 |

Data was collected on a single crystal
[a]Values in parentheses are for highest-resolution shell

describe a common class of V2p binding antibodies with a characteristic mode of antigen recognition.

**IGHV5-51 class V2p antibodies inhibit HIV-1 binding to α4β7**. RV144 vaccine elicited V2p antibodies have been shown to block the binding of the α4β7 integrin to V2 peptides[35]. Since the CAP228 antibodies targeted the same epitope as CH58, the extent to which they could also perform this function was assessed using two different binding assays (Fig. 3). Using a flow cytometry-based assay, a full length, fully glycosylated, clade C consensus (ConC) gp120 was used as the antigen. This was bound to integrin α4β7 transiently expressed on HEK293T cells, thereby avoiding the involvement of CD4. Both the RV144 mAb CH58 and the CAP228 V2p mAbs effectively inhibited the binding of α4β7 expressing HEK293T cells to ConC gp120 (Fig. 3a). This was comparable to inhibition by two antibodies (HP2/1 and Act-1) that bind directly to α4β7, which were used as positive controls. A non-V2 directed HIV-1 broadly neutralising antibody (VRC01) targeting the CD4 binding site of gp120, and an irrelevant respiratory syncytial virus (RSV) antibody (Palivizumab) were used as negative controls and failed to block α4β7 binding.

In a second fluorescence/colorimetric cell adhesion assay, the CAP228 mAbs were assessed for their ability to inhibit the binding of RPMI8866 cells naturally expressing α4β7 (but not CD4 or CCR5) to a cyclic V2 peptide from the RV144 vaccine strain 92TH023, or the clade A strain BG505 (Fig. 3b, c). At higher concentrations (0.5 μg mL$^{-1}$) inhibition by CAP228-16H or CH58 was comparable, but at four-fold lower concentrations CAP228-16H still showed substantial inhibition of α4β7 adhesion while CH58 lacked any blocking activity (Fig. 3b). When tested against the RV144 92TH023 vaccine strain V2 bearing the H173 immunotype, all the V2p antibodies (from CAP228 or RV144) exhibited integrin blocking activity at 0.5 μg mL$^{-1}$ (Fig. 3c–left graph), comparable to the α4β7 reactive mAb 2B4 used as a positive control. Similarly, except for RV144 mAb CH59, all the V2p mAbs also effectively inhibited binding of α4β7 to the BG505 V2 peptide with a Y173 immunotype (Fig. 3c–right graph). Interestingly, a V1V2 conformation dependent mAb (non-V2p) also isolated from CAP228 (9D) that is not reliant on residues K168/K169 failed to inhibit α4β7 integrin adhesion (Fig. 3c). Altogether these data suggest that the ability to block V2 interactions with the α4β7 integrin is exemplified by the IGHV5-51 V2p class mAbs, modulated by the V2 position 173

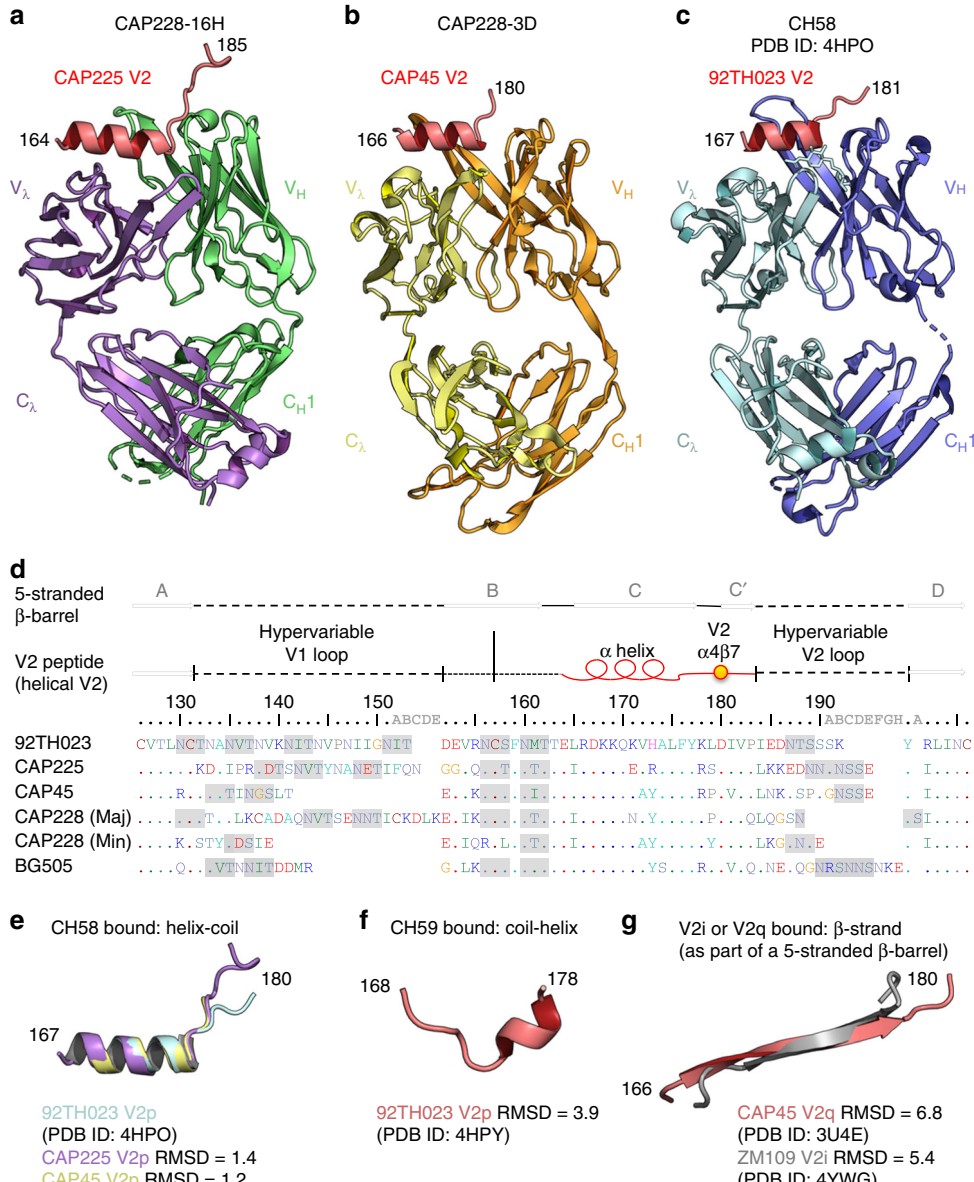

**Fig. 1** HIV-1 V2 peptides reproducibly sample the same helix-coil conformation. **a** A cartoon representation of the Fab region from V2p mAb CAP228-16H bound to a heterologous V1V2 domain from HIV-1 strain CAP225. Only V2 residues 164–185 were resolved, and are shown in red. The Fab heavy and light chains are coloured green and purple, respectively. **b** A cartoon representation of the Fab region from V2p mAb CAP228-3D bound to a short, heterologous V2 peptide (shown in red) from HIV-1 strain CAP45. The Fab heavy and light chains are coloured orange and yellow, respectively. **c** Previously determined cocrystal structure of CH58 bound to the RV144 vaccine strain V2 peptide. **d** A sequence alignment of the V1V2 domains from the HIV-1 strains for which crystal structures were determined in this study, as well as the dominant (Maj) and subdominant (Min) CAP228 cofounder viruses. The hypervariable loop regions of V1 and V2 are indicated with the dashed lines, while the conserved V2 helix-coil motif is labelled with a red cartoon representation. A gold sphere was used to mark the location of the primary α4β7 binding site, and any potential N-linked glycosylation sites are shaded grey. Above, arrows designate those regions of V1V2 that form the 5-stranded β-barrel bound by V2q/V2i mAbs. **e** Main chain superimposition of the V2 peptides (residues 167–180) bound by CAP228-16H (purple), CAP228-3D (yellow) and CH58 (light blue). The Cα root-mean-square deviation for each comparison with CH58 bound peptide is shown. **f** Previously determined structure of a CH59 bound V2 peptide (residues 168–178). **g** Previously determined structures of the V2 C-sheet in the context of the 5-stranded β-barrel (residues 166–180) when bound by PG9 (red) or 830 A (silver)

immunotype, and may represent an anti-viral function that is unique to V2p antibodies.

**Helix-coil epitopes on full length terminally constrained V1V2 scaffolds.** The isolation of CAP228 mAbs confirmed the existence of V2 peptide targeting B cell responses during HIV-1 infection, but it remains unclear whether these antibodies are stimulated by virion-free V2 peptide fragments, or by some other

antigenic form of V1V2. In the context of native gp120, the V1V2 region is constrained by two disulphide bonds, one at the base of the domain, and one internal bond that defines the V1 hypervariable loop. To study V2i and V2q epitopes in isolation, a monomeric but native-like V1V2 domain has previously been grafted onto various small protein scaffolds (usually based on PDBID 1FD6)[26,28,29,31,41]. These constrain the V1V2 N and C termini in a hairpin structure that facilitates appropriate disulphide bond formation. However, these same 1FD6-scaffolded

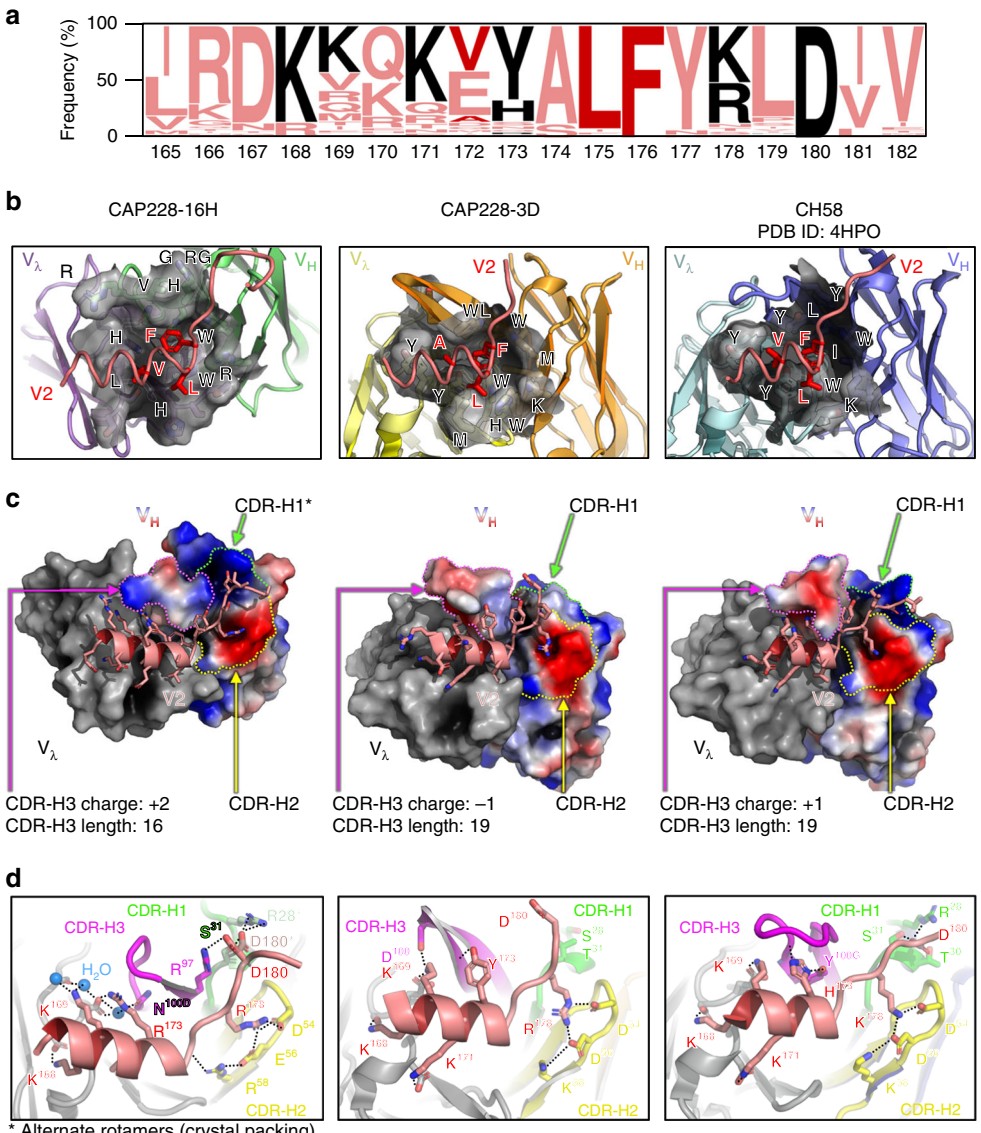

**Fig. 2** IGHV5-51 V2p mAbs define a binding class with a distinct mode of recognition. **a** Logogram of global V2 sequences in the HIV-1 LANL database, showing the relative amino acid frequencies (*y*-axis) at each position in V2 (*x*-axis, residues 165-182). Key hydrophobic amino acids bound by IGHV5-51 antibodies are highlighted in darker red, while other key contact residues are coloured black. **b** A cartoon representation of the three IGHV5-51 V2p Fab paratopes (coloured as in Fig. 1), with the central hydrophobic binding depression shown in surface view. Heavy and light chain residues that make up the hydrophobic binding area are indicated, and the side chains are shown for key buried V2 residues 172, 175 and 176. **c** The solvent accessible surfaces of the heavy and light chain variable domains from all three mAbs are shown and labelled. The light chains are coloured dark grey, while the heavy chains are coloured on a smoothed charge gradient with more electronegative regions in red, and more electropositive surfaces in blue. The heavy chain complementarity determining regions 1, 2 and 3 are encircled with green, yellow and pink dotted lines, respectively, and the CDR-H3 lengths and overall charges are indicated. V2 peptides are shown in stick and cartoon views. A single rotamer change to account for crystal packing is indicated by the asterisk. **d** Atomic level details of the interaction between all three IGHV5-51 mAb heavy chains and V2, with the CDRs coloured as in **c**. Hydrogen bonds are indicated with the dotted black lines, and key water molecules are shown with blue spheres. Alternative rotamer conformations for V1V2 residue D180 and CDR-H1 residue R28 (influenced by crystal packing in the CAP225 bound structure) are shown and labelled with asterisks

V1V2s were also bound by the CAP228 antibodies (and to a lesser extent CH58) in ELISA (Supplementary Fig. 1A). This binding was often better than what we have previously measured for the V2q antibody PG9, suggesting that like V2 peptide fragments, V1V2 scaffolds also present a conformationally heterogenous pool containing both α-helical and β-stranded forms of V2.

To characterise the structural differences of the entire V1V2 domain between these two states, we determined the cocrystal structure of CAP228-16H bound to a V1V2 scaffold from the autologous cofounder strain CAP228(Min) at 3.2 Å resolution (Fig. 4 and Table 1c). Despite the addition of a hairpin constraint

at the V1V2 termini (provided here by scaffolding onto 1FD6), the V2p epitope was efficiently captured by CAP228-16H in the same helix-coil configuration normally presented by short V2 peptides (Fig. 4a–shown in red). Both the primary α4β7 binding site at position D180 (Fig. 4–gold spheres) and the potential secondary α4β7 binding determinant that includes position K171 (Fig. 4–silver spheres) were colocalized in the CAP228-16H paratope. Electron density for V2 residues bound by CAP228-16H was clearly defined and matched well with the higher resolution V2 peptide cocrystal structures (Supplementary Fig. 3A). Overall, there were not any noteworthy additional

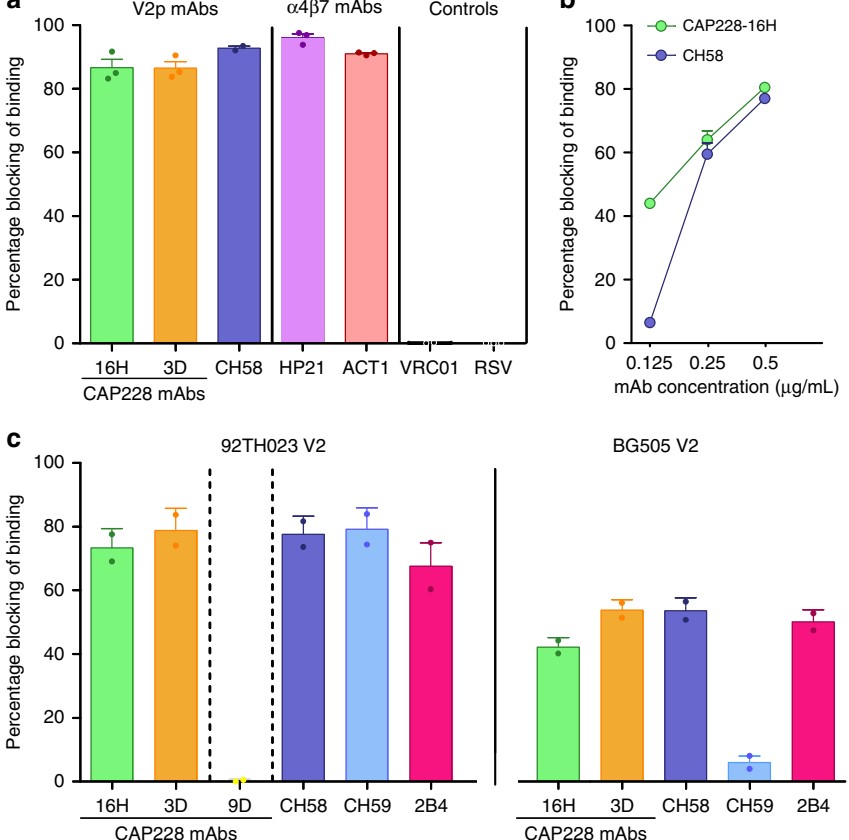

**Fig. 3** CAP228 antibodies efficiently inhibit α4β7 integrin binding. **a** Inhibition of ConC gp120 binding to α4β7 integrin transfected 293 T cells by V2p mAbs CAP228-16H (green), CAP228-3D (orange) or CH58 (blue). Percentage blocking (y-axis) was calculated from the difference in mean fluorescence intensity (MFI) when cell-gp120 complexes were made in the presence or absence of antibody. The α4β7 specific antibodies HP21 (purple) and ACT1 (red) were used as positive controls, while an irrelevant RSV antibody (palivizumab) and an HIV-1 CD4 binding site-specific antibody (VRC01) were used as the negative controls. **b** Ability of V2p antibodies CAP228-16H (green) or CH58 (blue) to block adhesion of α4β7 integrin expressing RPMI8866 cells to a cyclic 92TH023 peptide (residues 157–196) at three separate concentrations. **c** Inhibition of V2 peptide (strains 92TH023 or BG505) binding to α4β7 integrin expressing RPMI8866 cells by V2p mAbs CAP228-16H (green), CAP228-3D (orange), CH58 (blue) and CH59 (light blue) or the conformation specific non-V2p mAb CAP228-9D. The mAb 2B4 is specific for the α4 integrin and served as the positive control (pink). Data are represented as the result of up to three biological replicates used to calculate a mean

contacts between CAP228-16H and the newly resolved regions of CAP228(Min) V1V2, which buried a total surface area of ~980 Å$^2$ in the mAb paratope, similar to the ~940 Å$^2$ of surface area buried in the V2 peptide interaction (Supplementary Fig. 2B). As a result, the 1FD6 scaffold and V1V2 adjoining regions that were not bound by the antibody were much less ordered and suffered from higher B-factors, but the backbone Cα chain could still be reasonably traced through these regions (Supplementary Fig. 3B). Crystal packing enabled near complete visualisation of both V1 and the hypervariable loop region of V2 in one of two copies in the asymmetric unit, with the exception of three loop-apical V2 residues K185, G186 and N187 that were modelled in for completeness (Fig. 4 and Supplementary Fig. 3). In this complex V1 is pushed into a more helical fold (absent in the less constrained complex) that kinks at position 161 before leading into the V2p-associated α-helix bound by CAP228-16H, identifying potential V2 conformational switch residues in the N160 glycosylation sequon. The additionally resolved N- and C-terminal regions of V1V2 loosely encircled the CDR-H3 of CAP228-16H before inserting into the scaffold. Presumably this was also the case for the partially resolved cocrystal structure of CAP228-16H bound to the CAP225 V1V2 described above (Fig. 1a and supplementary Fig. 1B), however the heterologous

strain has significantly longer V1 and hypervariable V2 loops (Fig. 1d and Supplementary Fig. 1A), which would have introduced more flexibility and possibly contributed to the disorder of the heterologous complex. As with V2i/V2q antibodies that recognise the β-stranded form of V1V2, the V1 loop (Fig. 4–shown in white) and hypervariable loop region of V2 (Fig. 4–shown in dark blue) side chains in CAP228(Min) were not contacted by the CAP228-16H CDRs.

Visualised here, the V1 and V2 cystine bonds appeared correctly paired, despite substantial domain reorganisation (Fig. 4b–shown in yellow). The CAP228(Min) strain also has three potential N-linked glycosylation (PNG) sequons associated with V1V2, but at the current resolution no linked N-acetyl glucosamine moieties were visible in the cocrystal complex. However, asparagine side chains associated with each PNG sequon (N135 in V1, N187 in V2 and N160) were not buried in the antibody paratope and could conceivably be glycosylated. Similarly, V1V2 positions N130 and N156 are often glycosylated in other HIV-1 strains, and projected glycans at these positions also did not appear to clash directly with the CAP228-16H mode of recognition, though they would undoubtedly limit mAb accessibility to this site (Supplementary Fig. 3C—green/purple lines). Altogether these data suggest that in the context of full

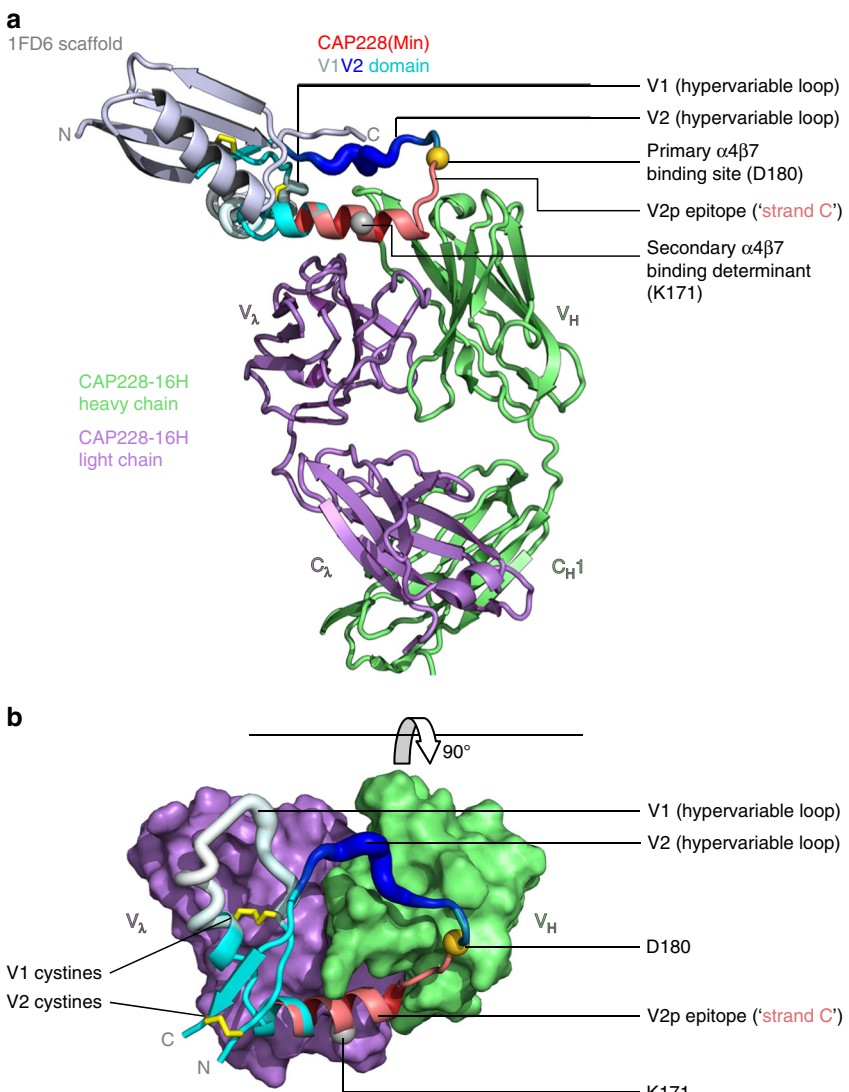

**Fig. 4** Cocrystal structure of a scaffolded, helical V1V2 domain bound to CAP228-16H. **a** The 1FD6 scaffold (metallic blue), autologous CAP228(Min) V1V2 domain (multicolour) and CAP228-16H Fab heavy and light chains (green and purple, respectively) are shown in cartoon view. V1V2 regions are labelled as hypervariable loop 1 (residues 132–156, white), hypervariable loop 2 (residues 183–196, dark blue), and C strand (residues 167–179, red), with the remainder coloured cyan. Internal V1V2 disulphide bonds are shown as yellow sticks, while the primary α4β7 binding site and potential secondary α4β7 binding determinant are indicated with the gold and silver spheres, respectively. **b** The view has been rotated 90°, and the 1FD6 scaffold (which would point 'out of the page') has been omitted to clearly show the interaction. The CAP228-16H paratope is shown in surface view, with the V1V2 domain still in cartoon representation, coloured as above

length V1V2, where the N and C termini are covalently associated, the V1V2 domain can fold into both the β-stranded and α-helical alternative conformations, the latter of which is bound by V2p antibodies.

**V2p mAbs bind to refolded V1V2 compatible with CD4-bound gp120.** To investigate the role of α-helical conformations of V1V2 on HIV-1 Env, we next compared the binding angles and interactions of V2p (CAP228-16H), V2q (PG9, PDB ID: 3U4E) and V2i (830 A, PDB ID: 4YWG) mAbs to full length V1V2, using the 1FD6-scaffold to orient the interaction (Fig. 5). Since V1V2 is equally constrained in both the 1FD6 scaffold and gp120, the scaffold could be used to approximate the location of gp160 relative to these V2 targeted antibodies (Fig. 5a, dashed line). In all three structures the location of the V1 hypervariable loop was equally oriented away from the equivalent location of gp160, or potential trimeric interfaces, by the internal V1V2 disulphide

bonds (Fig. 5a, b). Similarly, the hypervariable loop region of V2 was also positioned toward bulk solvent, and away from Env. In this way α-helical V1V2 remains compatible with the considerable variation in loop length and glycosylation seen between different HIV-1 strains.

Both V2p and V2q mAb epitopes share an overlapping region that includes a cationic patch (residues 168-171) in the α-helix or C-strand of V1V2, respectively (Fig. 5a–coloured red). This includes the potential secondary determinant of α4β7 binding (residues 170–173), not recognised by V2i mAbs (Fig. 5a, b–silver sphere). Similarly, V2p and V2i epitopes share a region C terminal to the cationic patch that overlaps with the primary α4β7 binding site (Fig. 5a, b–gold sphere), not bound by V2q mAbs. The ability of V2p mAbs to bind both the primary and potential secondary α4β7 binding determinants stems directly from the α-helical conformation of V1V2 which facilitates colocalization of these two regions. Despite large structural rearrangements within

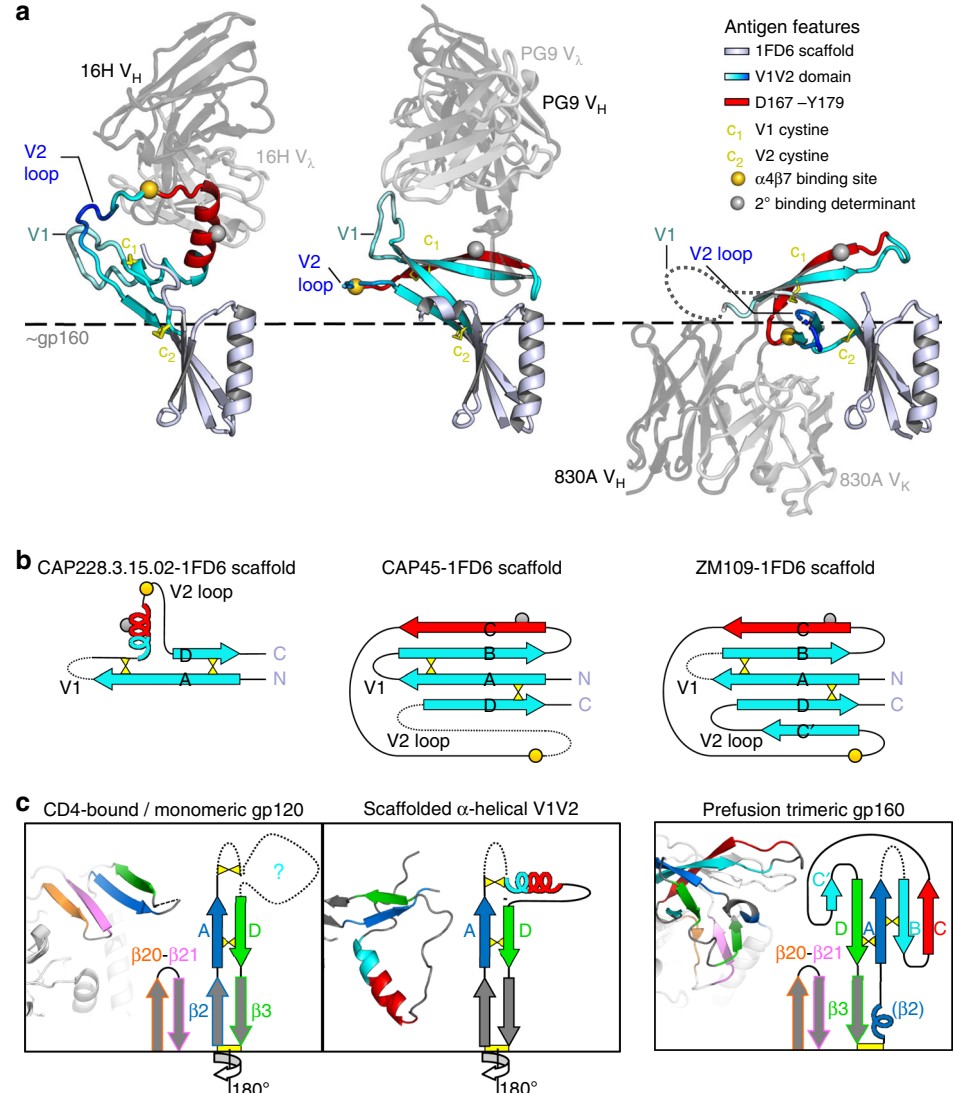

**Fig. 5** Comparisons of the V1V2 domain when bound by V2p, V2q and V2i mAbs. **a** A cartoon representation of CAP228-16H (V2p, left), PG9 (V2q, middle) and 830 A (V2i, right) bound to 1FD6 scaffolded V1V2. Only the Fab variable domains are shown for clarity, with the heavy and light chains coloured dark and light washed-out grey, respectively. All three structures are oriented with respect to the 1FD6 scaffold (metallic blue). Residues 167–179 that make up the V1V2 C-strand, or the IGHV5-51 bound α-helix, are shown in bright red, with the remaining V1V2 regions coloured as in Fig. 4. The V1V2 internal disulphide bonds are shown with yellow sticks, and α4β7 binding determinants are labelled with gold and silver spheres. A dashed line was used to indicate the approximate location for the rest of gp160 relative to full length V1V2. **b** Schematic of the V1V2 domain, showing the relative refolding of B- and C-strands, and repositioning of the A- and D-strands between the α-helical and β-stranded conformations of V1V2. The α4β7 binding determinants and disulphide bonds are labelled as in **a**. **c** Cartoon views and schematic representations of V1V2 in the CD4-bound or monomeric gp160 state (left panel), the V2p-bound α-helical state (middle panel), and the prefusion, entry-competent gp160 trimer (right panel). The β20 and β21 strands that make up one half of the bridging sheet are coloured orange and pink, respectively. The β2 strand of gp120 (or overlapping prefusion associated α-helical structure), and subsequent V1V2 A-strand region is coloured dark blue, while the V1V2 D-strand and subsequent β3 strand of gp120 is coloured green. The remaining V1V2 region is coloured as in **b**

V1V2 when comparing its α-helical and β-stranded forms, both the V2p and V2q epitopes were presented on a region of V1V2 that was most distal to the 1FD6-scaffold, and thus most distal to the approximate location for the rest of gp160 or the viral membrane. In contrast, V2i mAbs approach V1V2 from the opposite angle relative to V2p/V2q mAbs and would potentially clash with the rest of Env in this configuration[26,42]. Thus, unlike the epitopes recognised by V2i mAbs in the 5-stranded β-barrel, the α-helical conformation of V1V2 results in substantial exposure of V2p associated epitopes relative to the rest of gp160.

Another striking difference between β-stranded and α-helical conformations of V1V2 was the relative orientations of the V1V2

N and C termini (Fig. 5c). When V1V2 is folded into the 5-stranded β-barrel presented in the prefusion trimer[3,4], and recognised by V2q or V2i antibodies, its C terminal D-strand and β3-strand (Fig. 5c, highlighted in green) insert between its N terminal β2-strand and A-strand (Fig. 5c, highlighted in blue) and the β20-β21 hairpin, forming parallel strand interactions between β3 and β21 (Fig. 5c–right panel). Following CD4 engagement[6,7], or in the context of monomeric gp120[43], the β2-strand and β3-strand (and thus the A-strand and D-strand of V1V2) reposition 180° relative to each other, forming more stable antiparallel strand interactions between β2 and β21 known as the bridging sheet (Fig. 5c–left panel). After rearrangement of the V1V2

termini, the resulting structure of V1V2 in this CD4-bound orientation is unknown. Seen here, the α-helical form of V1V2 bound by CAP228-16H has similarly repositioned termini with respect to the spatial orientation of the D-strand of V1V2 relative to V1 (Fig. 5c–middle panel). Without the tertiary constraints imposed by the folding of the 1FD6 scaffold in the analogous β2-β3 region, the orientation of the A-strand and D-strand of α-helical V1V2 would be compatible with the formation of the bridging sheet in gp120. Altogether these data suggest that when released of the constraints imposed by native, trimeric, prefusion Env, V1V2 can alternatively fold into an α-helical form. This results in the colocalization and relatively high exposure of both the primary α4β7 binding site and secondary α4β7 binding determinant, as well as the epitopes recognised by V2p antibodies on CD4-bound/monomeric conformations of gp120.

**Helical V1V2 epitopes are presented on membrane associated gp160**. To provide more information about the physiological relevance of V2p mAbs, we next sought to confirm the presence of α-helical V1V2 within membrane associated gp160. Primary CD4$^+$ T cells were infected with replication competent viruses expressing the 92TH023 (RV144 vaccine strain) or CMU06 envelopes. Both envelope strains were from HIV-1 clade AE and had the H173 V2 immunotype important for CH58/CH59 binding. The infected CD4$^+$ T cells were then allowed to proliferate for several days before being stained with the RV144 V2p mAbs CH58 and CH59 (Fig. 6a). In contrast to the uninfected controls, a significant proportion of the infected cell groups were stained with the RV144 derived antibodies, confirming the existence of membrane associated α-helical V1V2 and V2p epitopes on the cell surfaces. To further extend these analyses, HIV-1 virus particles expressing the C1080 envelope were assessed for binding to V2p, V2i or V2q antibodies in ELISA (Fig. 6b). C1080 is an HIV-1 clade C strain expressing the globally common Y173 immunotype that stimulated both CAP228 V2p lineages. Consequently, both CAP228 antibody lineages bound strongly to the concentrated virion particles, equivalent to the binding of a gp120 specific HIV-1 antibody VRC01 used as the positive control. Similarly, the epitopes for V2i mAb 830 A or quaternary structure dependent V2q mAb PGT145 were also detected. PGT145 binding was relatively weaker than V2p mAb binding, consistent with the fact that 'closed' prefusion Env trimers are less abundant on virion surfaces[8]. Binding of the RV144 mAb CH59 (which is specific for the H173 immunotype) or an irrelevant influenza-specific antibody could not be detected. Thus, V2p mAb epitopes are readily detectable on both infected cell surfaces and HIV-1 virions and represent a potential mechanism through which V2p mAbs might exert an α4β7 blocking antiviral effect.

## Discussion

The α4β7 integrin is non-essential for the infection of CD4$^+$ T cells by HIV-1, but can serve as an adhesion receptor that is being increasingly shown to play an important role in HIV-1 pathogenesis[17,19,20]. Antibodies that block this interaction by targeting α4β7 can reduce infection or substantially affect disease outcome in acutely SIV infected non-human primates, while V1V2-targeted antibodies have been implicated as a correlate of reduced risk of HIV-1 infection in the RV144 vaccine trial[22–24]. Thus far, the mechanism whereby V2p antibodies might contribute to this effect against HIV-1 viruses has been unclear, but may include Fc-mediated effector functions[44]. Here, we use X-ray crystallography to define a class of V2p mAbs that all derive from a common IGHV5-51 heavy chain gene and are able to block the binding of HIV-1 to α4β7 by recognising a common helix-coil epitope. Additionally, we show that V2p helix-coil epitopes are

presented as alternatively folded forms of V1V2 which exist on infected cell surfaces and virion particles. These likely represent aberrantly misfolded, incompletely glycosylated, non-trimeric or CD4-triggered gp160 protomers that display a highly exposed α4β7 binding site. Overall, these data provide a structural explanation for how V2p antibodies may mediate antiviral activities.

V2p mAbs can be broadly categorised by the requirement for an anionic light chain motif, critical for the binding of V2 residues 169–173 (van Eeden, C., Wibmer, C. K. et al., manuscript submitted; and ref.[39]). Previous structural studies initially suggested that V2p mAbs recognise diverse conformations of V2[35]. In contrast, the monoclonal antibodies from HIV-1 infected donor CAP228 reveal a common V2 helix-coil conformation that is recognised by several different antibody lineages both from HIV-1 infection or vaccination. This helix-coil conformation was reproducibly adopted by monomeric V1V2 from several clade AE and clade C strains, independent of V1 or V2 loop length or levels of V1V2 glycosylation. Thus, despite some conformational plasticity, the common mode of antigen recognition described herein suggests that the V2 region adopts at least two reproducible folds: the 5-stranded β-barrel recognised by V2q and V2i antibodies[26,28,29,31], and the cyclical helix-coil conformation recognised by CH58-like IGHV5-51 class V2p antibodies. The functional significance of the β-barrel conformation has been known for some time, but an understanding of the role of helical V1V2 has been limited by the absence of V2p antibodies that contact appropriate immunotypes, particularly at position 173. The CAP228 antibodies have therefore provided an opportunity to explore this alternative conformation, and revealed that these helix-coil structures are present on infection relevant cell/virion membrane surfaces.

The IGHV5-51 class of V2p mAbs have a convergent 'hook and reel' mode of binding, where both heavy and light chain CDR2 loops are rigid, preconfigured hooks to catch the helix-coil form of V2, while the CDR1 loops then affinity mature to reel in the two α4β7 binding determinants (S28R-D180 or N31D-K171, respectively). The codon for position 28 spans two activation induced deaminase (AID enzyme) hotspots and is the most mutable amino acid in the VH5-51 gene, making R28 relatively easy to select through vaccination/infection. In addition to this mechanism of maturation, the presence of R97 in the CDR-H3 of CAP228-16H may have also contributed to D180 binding in the context of the initial CAP228 B cell activation event. Historically IGHV5-51 heavy chain antibodies paired with lambda light chains were identified as being preferentially selected for by the HIV-1 V3 loop antigen[45]. A comparison between the published crystal structures of IGHV5-51 antibodies bound to V3[46], and IGHV5-51 V2p mAbs from this study, revealed several overlapping features of V2p and V3 recognising antibodies (Supplementary Fig. 4). It is possible that the HIV-1 V3 loop is more antigenic, disfavouring the maturation of early cross-reactive IGHV5-51 antibodies to V2, or that virion associated α4β7 contributes toward hiding helical V1V2 from the immune system[18,23]. Alternatively, the ability of V2p mAbs to block viral interactions with the α4β7 integrin might drive viral escape mutations in Env that select against the further development of V2p binding antibodies. Non-neutralising antibodies to other epitopes in HIV-1 Env have similarly been shown to select for viral escape mutations[47], and contribute towards protection from other viral pathogens[48]. Further study is needed to determine the extent to which V2p antibodies might drive the selection of HIV-1 Env during infection.

The CAP228 antibodies were able to efficiently block the binding of V2 to α4β7 integrin. Importantly RV144 mAbs CH58 and CH59 were elicited by clade AE antigens with an H173

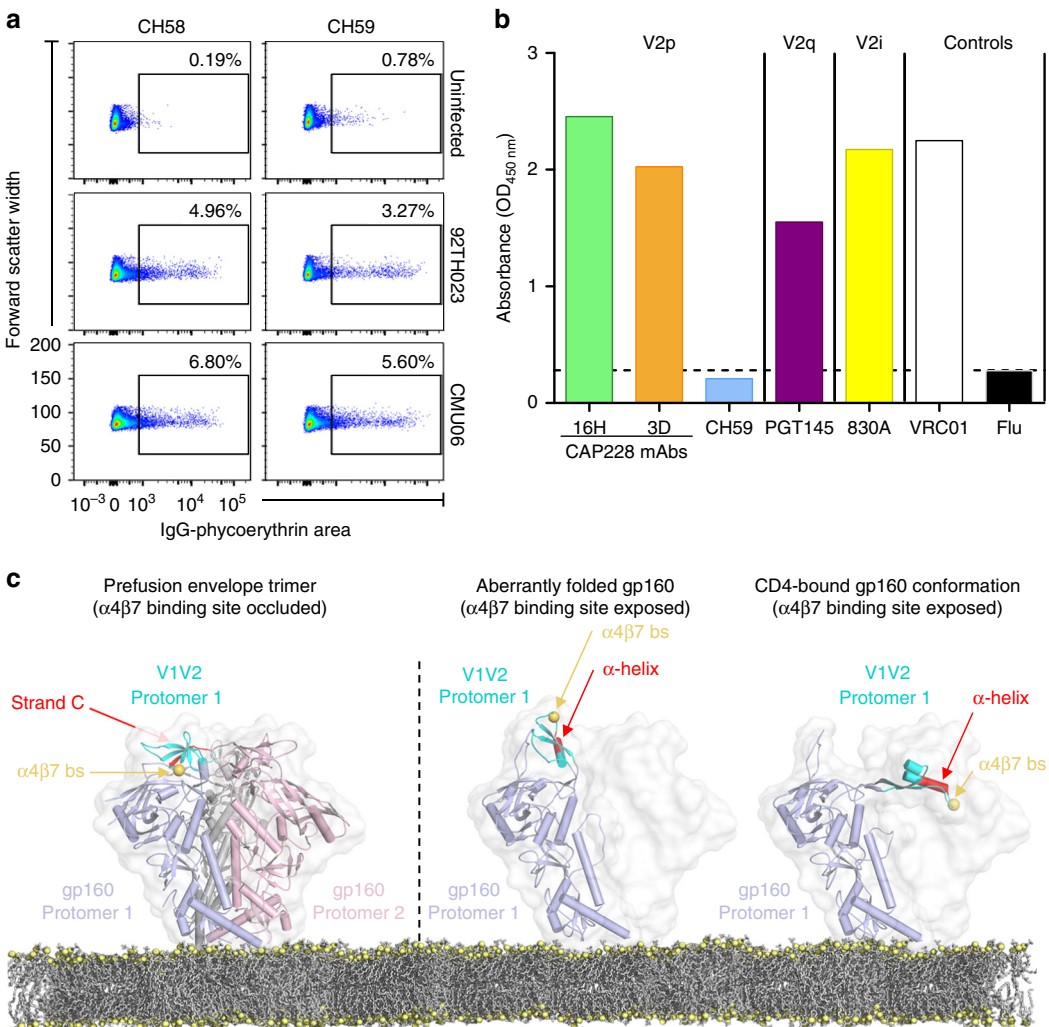

**Fig. 6** The α-helical form of V1V2 potentially engages the α4β7 integrin with high efficacy. **a** Cell surface staining of HIV-1 infected CD4+ T cells (strains 92TH023 and CMU06) using RV144 mAbs CH58 (left) and CH59 (right). An uninfected control (top panels) was included to estimate background and showed minimal (<1%) staining of cells. Plots are shown as forward scatter (cell size indicator) vs. level of phycoerythrin conjugated staining of either mAb. **b** Endpoint ELISA binding titres of V2p mAbs CAP228-16H (green), CAP228-3D (orange) or CH59 (light blue), quaternary structure dependent V2q mAb PGT145 (purple) or V2i mAb 830 A (yellow) to concentrated pseudovirus expressing the C1080 envelope. HIV-1 CD4 binding site-specific antibody VRC01, and an irrelevant Flu-specific antibody were used as positive or negative controls, respectively. **c** Models of the HIV-1 prefusion Env trimer (left), misfolded/refolded helical V1V2 (centre) or gp160 that has adopted the CD4-bound conformation (right) are shown. The latter two conformations of Env display highly exposed α4β7 binding sites (gold spheres) within helical V1V2, providing a potentially functional role for aberrant Env on the viral surface in α4β7 integrin binding and viral dissemination

immunotype, which is less globally prevalent, limiting their reactivity. This is particularly true for CH59, which fails to bind to most Y173 containing V2 antigens. The CAP228 antibodies are more tolerant of sequence variation at this position, recognising both common and rare V2 immunotypes, and therefore have a greater breadth of α4β7 blocking activity. Interestingly, while both V2i and V2p antibodies have been shown to mediate effective ADCC of HIV-1 infected cells (van Eeden, C., Wibmer, C. K. et al., manuscript submitted; and ref [44]), the conformation dependent antibody CAP228-9D could not block the binding of V2 to α4β7. Similarly, the V2i mAb 830 A cannot prevent α4β7 binding, despite interacting directly with the primary integrin binding tripeptide motif[49]. These data suggest that some of the antiviral functions performed by V2p antibodies are V2 conformation specific, and therefore cannot be recapitulated by β-strand preferring V2i or V2q antibodies. Thus far, helical V2p epitopes have only been characterised in the context of short V2 peptides[35]. Computational V1V2 modelling and Nuclear

Magnetic Resonance studies of slightly longer, unconstrained V2 peptides suggested that the conserved portion of V2 has a propensity for adopting an alternate α-helical fold, but it remained unclear how this conformation was related to full length Env[27,38]. The 1FD6 scaffold used here to characterise the entire N-terminally and C-terminally constrained V1V2 domain in the V2p helix-coil reveal how an alternatively folded α-helical V1V2, that was not the result of aberrant disulphide formation, forms V2p epitopes through a 180° relocation of the V2 exit loop (strand D) relative to V1 (and strand A). We hypothesise that the refolded α-helical form of V1V2 is more compatible with the CD4-bound conformation of gp120 (and the formation of the bridging sheet), which has been freed of some oligomeric constraint. This circular conformation likely resembles the structures adopted by cyclic V2 peptides (C157-C196) commonly used to characterise V2p antibody responses. The 16-19 amino acid CDR-H3 lengths normally exhibited by IGHV5-51 V2p antibodies may be explained by their position at the centre of this

cyclic epitope, where they contribute to removing internal solvent molecules and reducing entropy to stabilise the interaction. Refolding of V1V2 is inhibited in-part by the steric constraints imposed by glycosylation, and it has been shown that deglycosylated V1V2 binds far better to $\alpha 4\beta 7$[50,51]. However, the V1V2 scaffolded antigens, viral particles, and infected cells described here were all bound by V2p antibodies without any deglycosylation required, suggesting an inherent propensity for monomeric V1V2 to adopt this conformation.

In the prefusion closed conformation of gp160, and its associated 5-stranded $\beta$-barrel fold of V1V2, the $\alpha 4\beta 7$ binding site is largely occluded from antibody binding (Fig. 6c, left). These data are consistent with the observations that V2p antibodies only neutralise phenotypically tier-1 viruses that have an open conformation with more exposed V2 and V3 loops. However, V2q, V2i and V2p antibodies all effectively bind to membrane associated gp160 both on HIV-1 infected cells and virion surfaces. When modelled onto membrane-bound, CD4-triggered or otherwise aberrant gp160 monomers, the $\alpha$-helical fold of V1V2 has a highly exposed $\alpha 4\beta 7$ binding site and facilitates colocalization of the primary and secondary $\alpha 4\beta 7$ binding determinants (Fig. 6c, middle and right). Altogether these data suggest that aberrant forms of gp160 protomers still anchored on the surfaces of HIV-1 virions or infected cells preferentially adopt an $\alpha$-helical V2 conformation that is conserved across HIV-1 clades, and efficiently displays the $\alpha 4\beta 7$ integrin binding site. The ability of IGHV5-51 V2p antibodies to bind this alternative form of Env and block the $\alpha 4\beta 7$ interaction provides one mechanism through which these non-neutralising antibodies contributed to the effectiveness of an HIV-1 vaccine[52]. Thus, we propose a functional role for aberrant forms of Env in early HIV-1 pathogenesis, which has important implications for V1V2-targeted vaccine design.

## Methods

**Protein expression and purification.** Monoclonal antibodies were expressed as IgG1 subclass from separate heavy and light chain encoding CMV/R plasmids[53] (backbone vector sequences can be obtained from the NIH AIDS reagent repository https://www.aidsreagent.org/qsearch.cfm), engineered to include an HRV-3C protease cleavage site (GLEVLFQGP) between the Fab and Fc fragments, in Expi293F cells (ThermoFisher Scientific) using PEI-MAX 40,000 (Polysciences). Cell culture supernatants were harvested seven days after transfection and purified using protein A (BioVision). The antibodies were digested with HRV-3C (Merck Millipore) at 25 °C for four hours to release the Fab fragments. Fab-peptide complexes were then incubated with a 10-fold molar excess of linear V2 peptides (synthesised by Genscript) before purification by size exclusion chromatography using a superdex 200 column (GE Healthcare).

ConC gp120 was similarly expressed in adherent HEK293T/17 cells (ATCC, CRL-11268) seeded into hyperflasks (Corning), and purified from cell culture supernatants by sequential *Galanthus nivalis* affinity (Sigma) and Q sepharose ion exchange (ThermoFisher Scientific) chromatography.

The V1V2 scaffolded proteins were expressed in HEK293S cells genetically altered to lack N-acetylglucosaminyltransferase I (ATCC, CRL-3022), in suspension culture for seven days using 293Freestyle media supplemented with 2% Foetal bovine serum (ThermoFisher Scientific), and fed with Hyclone™ SFM4HEK293 (GE Healthcare) 24 h post-transfection. The V1V2 scaffolds were captured from the cell culture supernatant with Ni sepharose™ excel (GE Healthcare), and further purified using size exclusion chromatography. Following this initial purification, V1V2 scaffolds were captured with CAP228 mAbs bound to protein A to obtain the antigen-antibody complexes. These Fab-scaffold complexes were deglycosylated on-column with Endo H (New England BioLabs), eluted following HRV-3C digestion overnight at 4 °C (to cleave off their Fc fragments and any His8 tags), and then further purified using size exclusion chromatography on a superdex 200 column (GE Healthcare). All cell lines were negative for mycoplasma testing.

**ELISAs.** V1V2 protein scaffolds were directly coated onto high binding ELISA plates (Corning) overnight at 4 °C at 4 μg mL$^{-1}$. V2 peptides were C-terminally biotinylated and coated onto streptavidin ELISA plates. The plates with blocked with PBS (0.05% polysorbate 20, 5% milk powder) for 2 h, before being probed sequentially with five-fold serially diluted monoclonal antibodies (starting at 10 μg mL$^{-1}$) and an anti-Fc HRP conjugate used a 1:10,000 dilution (Sigma, Cat.

No. A0170-10ML) in PBS (0.05% polysorbate 20, 5% milk powder) for 1 h each. The plates were developed using 1-step ultra TMB-ELISA solution (ThermoFisher Scientific) followed by 1 M sulphuric acid and read at 450 nm.

Envelope pseudotyped viruses (PSV) were grown in HEK293T/17 cells by co-transfecting the C1080 envelope plasmid and a pSG3ΔEnv backbone. Cell culture supernatants were collected after two days and filtered through 0.45 μm. PSVs were purified through a 20% sucrose cushion at 40,000×$g$, resuspended in PBS, and coated directly onto high binding ELISA plates. Plates were washed and probed as above in PBS (10% FBS, 2–4% BSA).

**Macromolecular X-ray crystallography.** Fab-peptide and Fab-V1V2 scaffold complexes were concentrated to 10–20 mg mL$^{-1}$, aliquoted, and flash frozen in liquid nitrogen. Thawed aliquots were hand-screened by sitting drop vapour diffusion in a humidified chamber using a 96-well deep block plate format (corning), that included the preformulated Wizard Precipitant Synergy and Wizard Classic 1–4 crystallisation screens (Rigaku). Crystal hits were identified in 1 μL drops containing 50% mother liquor, and further hand-optimised in 15-well hanging drop plates at 25 °C. Crystals of the CAP228-3D Fab complexed with a CAP45 V2 peptide (gp120 residues 164–182) were initially obtained in drops containing 0.1 M imidazole HCl (pH6.5), 20% PEG8000, 3% 2-methyl-2,4-pentanediol (MPD) or 0.1 M HEPES NaOH (pH7.5), 20% PEG8000, 0.2 M ammonium sulphate, 10% isopropanol. Larger crystals were obtained by mixing both conditions in a 1:3 ratio, supplemented with 25% MPD as a cryoprotectant. Crystals of CAP228-16H in complex with the CAP225-V1V2 1FD6-scaffold were obtained in 0.1 M HEPES NaOH (pH7.5), 20% PEG8000, 0.2 M ammonium sulphate, 10% isopropanol and cryoprotected with 20% glycerol, while crystals of CAP228-16H in complex with the autologous CAP228-V1V2 1FD6-scaffold were obtained in 0.1 M Tris HCl (pH8.5), 5% PEG4000, 2 M sodium chloride, further optimised to include 10% butanol, and cryoprotected in 25% 2,3-Butanediol. Diffraction data was collected at SER-CAT ID-22 beamline (Advanced Photon Source, Argonne National Laboratory), at a wavelength of 1.00 Å, 100 K and processed with HKL2000. Model building and refinement was handled with winCOOT v0.8 and PHENIX v1.9-1692, using 5% of the data as an R-free cross validation test set, and with refined hydrogens to minimise clashes.

**Data analysis.** All sequence alignments were created in BioEdit. HIV-1 Env sequences were numbered according to the HXB2 convention[54], while antibody sequences were numbered by the Kabat convention[55]. All graphs were generated with Graphpad, while structure figures were created in PyMol, and the RMSD, approximate charge, and BSA were estimated using PyMol and PISA.

**Integrin $\alpha 4\beta 7$ blocking assays.** The flow cytometry based $\alpha 4\beta 7$ blocking assay was adapted from previously described methodologies[56,57]. HEK293T/17 cells were co-transfected with $\alpha 4$ and $\beta 7$ encoding plasmids (Origene), cultured for two days, and then gently moved into solution with 1 mM EDTA/PBS. Co-expression of $\alpha 4\beta 7$ was confirmed by flow cytometric staining with CD49d (anti-$\alpha 4$) phycoerythrin (eBioscience, Cat. No. 12-0492-81) and anti-human/mouse integrin $\beta 7$ fluorescein isothiocyanate. Cells were washed into HBS (10 mM HEPES, 150 mM NaCl, 1 mM MnCl$_2$, 100 μM CaCl$_2$) in order to activate the integrin, and then incubated with or without mAb (five-fold titrations starting at 20 μg mL$^{-1}$) for 15 min at 4 °C, followed by biotinylated ConC gp120 (5 μg mL$^{-1}$) for 25 min. Cells were washed 3 times in HBS buffer, and stained with phycoerythrin linked streptavidin for 30 min in order to detect gp120.

**Cell based $\alpha 4\beta 7$ adhesion assay.** This assay was adapted from previously described work[58] as follows. Flat bottom 96-well polypropylene plates (Greiner Bio-One) were coated in triplicate overnight at 4 °C with NeutrAvidin. The NeutrAvidin-coated plates were then incubated with a biotinylated cyclic V2 peptide from HIV-1 strains BG505 or 92TH023 (5 μg mL$^{-1}$ in bicarbonate buffer) for 1 h at 37 °C. The supernatants were then removed and the plates blocked with 25 mM Tris, 2.7 mM potassium chloride, 150 mM sodium chloride, 0.5% BSA, 4 mM manganese chloride, pH 7.2 for 1 h at 37 °C. The solution was discarded and plates were manually washed 4 times with the same blocking buffer. After this 2 × 10$^5$ RPMI8866 $\alpha 4\beta 7^+$ cells, which had been pre-incubated for 40 min at 37 °C in 50 μL sample buffer in the absence or presence of anti-V2 or anti-$\alpha 4$ mAb at 10 μg mL$^{-1}$, were added at 37 °C (5% CO2) for 1 h. The wells were then washed 5 times with PBS followed by the addition of 100 μl of RPMI-1640 containing 1% FBS, 1% pen/strep/glutamine, 25 mM HEPES with 10 μL/well of AlamarBlue® dye. Fluorescence (excitation 560 nm and emission 590 nm) was measured immediately after the addition of the AlamarBlue® dye for 8 h.

**Cell surface staining.** CD4$^+$ T-cells were isolated from fresh, healthy donor PBMCs by negative selection using magnetic beads (StemCell Technologies). Cells were cultured in RPMI-1660 media supplemented with 10% FBS and 2% Penicillin/ Streptomycin/L-Glutamine and activated with OKT3, IL-2 (20 U mL$^{-1}$) and retinoic acid (10 nM) for 4 days. Cells were then infected or mock-infected with HIV-1 strains CMU06, TH023, or a media-only control. Viral stocks were generated as infectious molecular clones using an NL4.3 HIV-1 backbone and a separately cloned Env gene from either CMU06 (Genbank accession #AY669771)

or TH023 (Genbank accession #KU562843) transfected into HEK293T cells and passaged once through activated PBMCs. Infected CD4+ T-cells were stained six days post infection using $2 \times 10^5$ cells and 1ug of either CH58 or CH59 anti-gp120 mAb per test. Data were acquired using a BD FACS Canto II and all flow data were analysed using FlowJo.

## Data availability

The data generated from this study are available online in the RCSB Protein Data Bank (https://www.rcsb.org/) with the accession codes 6FY1, 6FY2 and 6FY3. The authors declare that all other data supporting the findings of this study are available within the article and its Supplementary Information files, or are available from the authors upon request.

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

## Acknowledgements

We are incredibly grateful to Prof. B. Trevor Sewell and his staff and students at the Structural Biology Research Unit (IIDMM, UCT), Cape Town, for use of equipment and invaluable assistance preparing our crystal shipment. We would also like to thank the Southeast Regional Collaborative Access Team (SER-CAT) at the Advanced Photon Source, Argonne National Laboratory, Chicago, for assistance at beamline 22-ID where all data was collected. Use of the Advanced Photon Source was supported by the U. S. Department of Energy, Office of Science, Office of Basic Energy Sciences, under Contract No. W-31-109-Eng-38. We acknowledge the contributions of CAPRISA participants and staff that made these types of studies possible, as well as those of Dr. Charmaine van Eeden, Dr. Bronwen E. Lambson, Dr. Nono N. Mkhize and Dr. Jinal N. Bhiman who all contributed towards the isolation of the CAP228 monoclonal antibodies. This project was supported by an NIH R01 (AI104387-01), the Medical Research Council of South Africa, the NHLS Research Trust, and the Poliomyelitis Research Foundation. P.L.M. is supported by the South African Research Chairs Initiative of the Department of Science and Technology and National Research Foundation of South Africa (Grant No 98341).

## Author contributions

C.K.W., P.L.M. and L.M. conceived and designed the project. C.K.W, S.I.R., J.Y., C.C., J. A., P.L.M. and L.M. conceived and designed experiments and contributed to data interpretation. C.K.W. performed all X-ray crystallographic or ELISA experiments and data analyses. S.I.R., J.Y. and C.C. performed α4β7 blocking and flow cytometric assays and subsequent data analyses. C.K.W. and L.M. wrote the paper, on which all authors commented.

## Additional information

**Competing interests:** The authors declare no competing interests.

