## [Peer Review File · Nature Communications]

Common helical V1V2 conformations of HIV-1 Envelope exposes the $\alpha 4\beta 7$ binding site on intact virions

Constantinos Kurt Wibmer^{1,2,*},, Simone I. Richardson^{1,2}, Jason Yolitz^{3,4}, Claudia Cicala³, James Arthos³, Penny L. Moore^{1,2,5}, Lynn Morris^{1,2,5}

1 Centre for HIV and STIs, National Institute for Communicable Diseases (NICD), of the National Health Laboratory Service (NHLS), Johannesburg, 2131, South Africa; **2** Faculty of Health Sciences, University of the Witwatersrand, Johannesburg, 2000, South Africa; **3** Laboratory of Immunoregulation, National Institute of Allergy and Infectious Diseases, National Institutes of Health, Bethesda, MD 20892, USA; **4** Institutes of Health–Johns Hopkins University Graduate Partnership Program, National Institutes of Health, Bethesda, MD 20892, USA; **5** Centre for the AIDS Programme of Research in South Africa (CAPRISA), University of KwaZulu-Natal, Durban, 4041, South Africa

*Current address: The Scripps Research Institute, La Jolla, USA

Corresponding Author: c.k.wibmer@gmail.com

A

B

Supplementary Figure 1: Antigen selection for crystallization studies

(A) Annotated sequence alignment and properties of candidate antigens screen by ELISA for CAP228-16H crystallization studies. For each strain used, the entire V1V2 sequence, number of potential N-linked glycosylation sites (PNGs), V1 and hypervariable region of V2 loop lengths, and relative binding in ELISA (scored as + poor, ++ moderate, +++ good, or ++++ very good) is shown. Successful crystallization is indicated with a Y. The titration curves for CAP228-16H and PG9 against each of the V1V2 antigens is shown, with mAb concentration on the x-axis vs. absorbance on the y-axis. * V1 loop length was defined between cystine flanking residues from position 132 – 156 of Env. ** The hypervariable region of V2 was defined as position 182 – 190 of Env. (B) Crystal lattice of CAP228-16H (shown in green and purple) bound to the heterologous V1V2 domain from HIV-1 strain CAP225 (shown in yellow). The approximate location of the disordered 1FD6 scaffold is indicated.

Supplementary Figure 2

A

CAP228-16H in complex with CAP45 V2 peptide (PDB ID 6FY0)

CAP228-3D in complex with CAP45 V2 peptide

B

Relative buried surface area of V2

Key:

S – salt bridge

H – side chain hydrogen bond

h – main chain hydrogen bond

CAP228-16H Heavy and Light chain contacts
 CAP228-3D Heavy and Light chain contacts
 CH58 Heavy and Light chain contacts

Supplementary Figure 2: Light chain interactions and relative buried surface area for V2 peptide complexes

The solvent accessible surfaces of the heavy and light chain variable domains from CAP228-16H and CAP228-3D bound to the CAP45 V2 peptide are shown and labelled. The heavy chains were coloured light grey, while the light chains were coloured on a smoothed charge gradient with more electronegative regions in red, and more electropositive surfaces in blue. The light chain CDR-L2 associated electronegative patch is indicated (yellow arrow). V2 peptides are shown in stick and cartoon views, and the Y173 residue is indicated. The relative buried surface area for each amino acid in V2 (position 164 to 182) was calculated as the percentage of area normally accessible to solvent that is instead buried in the paratopes of the mAbs CAP228-16H (top two rows), CAP228-3D (middle row), or CH58 (bottom row) respectively. This percentage of buried surface area is shown on individual doughnut pie charts for each V2 position, where the percentage of surface area of each amino acid that is buried in the CAP228-16H heavy or light chain is coloured purple or green respectively, the percentage that is buried in the CAP228-3D heavy or light chain is shown in orange or yellow, and the percentage that is buried in the CH58 heavy or light chain is shown in blue or cyan respectively. Residues involved in hydrogen bonds (H/h) or salt bridges (S) are labelled. For each pie chart, the percentage surface area that remains solvent accessible for each amino acid in V2 (not buried in the antibody paratope) is coloured grey.

A

B

C

Supplementary Figure 3: Modelling V2p-Fab interactions

(A) Cartoon model showing the potential N-linked glycosylation sequons N135 (black), N187 (pink), and N160 (olive green) associated with the autologous CAP228 α -helical V1V2 shown in Figure 4. Commonly glycosylated sites at N156 (green) and N130 (purple) are also shown. The remaining structure are labelled as in Figure 4. (B) 2Fo-Fc density maps are shown for the V2p epitope region (residues 166-182) of the CAP228(Min) V1V2 when compared to the CAP45 peptide bound structure (PDB ID 6FY0). (C) 2Fo-Fc density maps are shown for the V1 loop and the hypervariable loop region of V2 from the CAP228(Min) V1V2 from each of the two complexes present in the asymmetric unit.

Crystal structure of IGHV5-51 mAb 4025 bound to a clade A consensus V3
PDB ID: 3UJJ

Supplementary Figure 4: IGHV5-51/Lambda interactions with V3

Atomic details of the interaction between IGHV5-51 mAbs with V3 peptides. HIV-1 V3 residues I307, I309, and F317/Y318 are similarly buried in the hydrophobic depression created between the heavy and light chain interface (left panel). The VH5-51 CDR-H2 anionic patch that clasps K/R178 in V2 forms a strong salt bridge with K305 in V3, and IGHV5-51 residue K/R58 similarly hydrogen bonds with the V3 peptide backbone (right panel). In addition, V3 antibodies that use the same ED-motif encoding light chains as V2p antibodies also use this motif to interact with R308 in V3 and can exhibit similar affinity maturation in the CDR-L1, where encoded residues interact with V3 at R/Q315, or V2 at K/Q171.